# Ultrafast magnetization enhancement via the dynamic spin-filter effect of type-II Weyl nodes in a kagome ferromagnet

Xianyang Lu [1,2,3,12], Zhiyong Lin [4,5,12], Hanqi Pi [6,7,8,12], Tan Zhang[9], Guanqi Li[10], Yuting Gong[2,3], Yu Yan[2,3], Xuezhong Ruan[2,3], Yao Li[2,3], Hui Zhang[4,5], Lin Li [4,5], Liang He [2,3], Jing Wu[10,11] ✉, Rong Zhang [3], Hongming Weng [6,7,8] ✉, Changgan Zeng [4,5] ✉ & Yongbing Xu [1,2,3,11] ✉

The magnetic type-II Weyl semimetal (MWSM) $Co_3Sn_2S_2$ has recently been found to host a variety of remarkable phenomena including surface Fermi-arcs, giant anomalous Hall effect, and negative flat band magnetism. However, the dynamic magnetic properties remain relatively unexplored. Here, we investigate the ultrafast spin dynamics of $Co_3Sn_2S_2$ crystal using time-resolved magneto-optical Kerr effect and reflectivity spectroscopies. We observe a transient magnetization behavior, consisting of spin-flipping dominated fast demagnetization, slow demagnetization due to overall half-metallic electronic structures, and an unexpected ultrafast magnetization enhancement lasting hundreds of picoseconds upon femtosecond laser excitation. By combining temperature-, pump fluence-, and pump polarization-dependent measurements, we unambiguously demonstrate the correlation between the ultrafast magnetization enhancement and the Weyl nodes. Our theoretical modelling suggests that the excited electrons are spin-polarized when relaxing, leading to the enhanced spin-up density of states near the Fermi level and the consequently unusual magnetization enhancement. Our results reveal the unique role of the Weyl properties of $Co_3Sn_2S_2$ in femtosecond laser-induced spin dynamics.

Weyl semimetals have attracted lots of attention due to their topological behaviors of the quantum electronic states and the correlated exotic physical properties[1–24], hosting emergent Weyl fermions in the bulk and Fermi arc surface states that connect the Weyl nodes of opposite chirality. Type-I Weyl semimetals were firstly predicted and experimentally verified in TaAs and other related compounds, which host a point-like Fermi surface at the Weyl nodes as shown in Fig. 1a[16–19,23]. Type-II Weyl semimetals are a class of materials with topologically protected Weyl nodes as the contact point between electron and hole pockets[25]. As shown in Fig. 1b, these Weyl nodes are realized due to the overtilted Weyl cones in momentum space where the valence and conduction bands touch. Recently, the magnetic Weyl

semimetals (MWSM) with time-reversal symmetry (TRS)-breaking crystals were discovered. This combination of magnetic ordering and Fermi nodes is particularly attractive for spintronic studies and applications[1–3,15].

Electron transition or relaxation around the Weyl nodes is key to determining the physical properties in Weyl semimetals under low-energy excitation. However, the role of Weyl nodes in electron and magnetization dynamics in the MWSMs with type-II Weyl nodes remains elusive. As shown in Fig. 1c (type-I) and 1d (type-II), the Weyl nodes in MWSMs are formed by the crossing of a pair of spin-up and spin-down bands. The in-plane projection of spins $(k_x, k_y)$ is aligned due to spin-momentum locking, while the out-of-plane projection

**Fig. 1 | Spin filter effect in Type-II MWSM, Crystal and electronic structures and the measured transient Kerr rotation of $Co_3Sn_2S_2$. a** Type-I Weyl node with a point-like Fermi surface. **b** Type-II Weyl node as the contact point between electron and hole pockets. spin-dependent electron-electron scattering in (**c**) gapless type-I Weyl node, **d** type-II Weyl node, **e** gapped type-I Weyl node and **f** gapped type-II Weyl node. **g** Unit cell in a hexagonal structure. The kagome lattice is formed by the cobalt atoms, of which the magnetic moments are along the c axis. **h** Zero field cool/Field cool measurements on temperature dependence of magnetization. The Curie temperature is 170 K. **i** Schematic illustration of the pump-probe experimental setup. **j** The measured transient Kerr rotation of $Co_3Sn_2S_2$ excited by a pump pulse with various fluence from 0.14 to 2.26 mJ/cm² at 10 K. Circles are experimental data and the solid lines are the fitted curves. **k** Illustration of the different dynamic components: A1 represents the magnetization enhancement process; A2 and A3 represent the fast and slow demagnetization processes, respectively; A4 represents the initial peak around zero delay.

($k_z$) of spins is determined by bands' polarization. It is worth noting that the dominant role of the electron-electron relaxation is experimentally observed in the type-II Weyl semimetal MoTe₂[26]. In MWSMs, the local inter-band transition is a spin-flipping process that requires electron-magnon or electron-phonon scatterings, of which the scattering rate could be significantly smaller than the electron-electron scattering rate. Therefore, we will only consider intra-band electron-electron relaxation when discussing electron (spin) relaxation across Weyl nodes in MWSMs under low-energy excitation.

In type-I MWSMs (gapless as shown in Fig. 1c, gapped as shown in Fig. 1e, the local symmetry of the spin-up and spin-down bands results in identical intra-band electron-electron relaxation rate and inter-band transition rate for both spin channels. Therefore, no collective spin accumulation is expected to form in type-I MWSMs during electron relaxation. On the other hand, in type-II MWSMs (Fig. 1d), spin-dependent electron relaxation is expected. Within a free electron model, the electron-electron scattering rate increases quadratically with energy relative to the Fermi level, as well as with a finite electron temperature[27,28]:

$$\tau_{ee}^{-1}(\varepsilon, T_e) \approx \frac{D_e}{\hbar}\left[(\varepsilon - \varepsilon_F)^2 + (\pi k_B T_e)^2\right], \quad (1)$$

where the proportionality $D_e$ can be treated as a constant for $|\varepsilon - \varepsilon_F| \ll \varepsilon_F$ and $T_e \ll \varepsilon_F/k_B$. Therefore, when the electrons are moderately excited with a finite equilibrium temperature, the electron-electron relaxation rates near the Weyl nodes would differ between the spin-up and spin-down electrons due to the overtilted Weyl cones. The local energy difference between the two spin-polarized bands can "accelerate" one of the spin polarized electron relaxation across the Weyl nodes, leading to a spin accumulation underneath the Weyl nodes. If this spin accumulation in the momentum space lasts long enough compared to the exchange coupling time (for instance, due to local half-metallicity band structure below the Weyl nodes blocking the spin-flipping scattering), it is proposed that a collective macroscopic magnetic ordering can form[29,30]. In the case shown in Fig. 1d, this magnetic ordering would manifest as a magnetization enhancement

because the spin-up polarization dominates below the Weyl nodes. This proposed spin filter effect only exist in gapless type-II Weyl nodes. If the Weyl nodes are gapped (Fig. 1f), the electron-electron scattering cannot dominate the process due to the bottleneck effect, and the local spin filter effect would no longer exist.

To verify the proposed physical picture, we choose the type-II MWSM $Co_3Sn_2S_2$ crystal to investigate the spin dynamics upon femtosecond laser pulse excitation. Ferromagnetic $Co_3Sn_2S_2$, with a kagome lattice structure, is one of the most explored MWSMs and serves as an ideal material platform for realizing rich and unusual physical phenomena. Giant anomalous Hall effect[15,31] and magneto-optical responses[4] are observed, arising from the considerably enhanced Berry curvature of its topological electronic structure. Electronic correlations are indicated to play a role in $Co_3Sn_2S_2$, resulting in a flattened band and correlated optical spectra[7]. Additionally, an unusual negative orbital moment is observed by scanning tunneling microscopy/spectroscopy (STM/S) in a kinetically frustrated kagome flat band[3]. It is found that the local disorders in $Co_3Sn_2S_2$ can induce elevated intrinsic anomalous Hall conductance[5] and spin-orbit polaron[6]. However, while these studies focus on static properties, dynamic magnetic properties such as the ultrafast magnetization manipulation remain elusive.

It has recently been demonstrated that varying magnetization can significantly shift the electronic band[8] and even lead to the annihilation of Weyl nodes and opening of a gap[12]. These results show that the band structure of MWSM $Co_3Sn_2S_2$, which contains Weyl nodes, can be effectively tuned by magnetization. Furthermore, half-metallic ferromagnetic behavior with a band gap in the spin-minority direction has been theoretically and experimentally demonstrated for $Co_3Sn_2S_2$[32,33]. Also, the demagnetization time upon femtosecond laser excitation of half-metals with high spin polarization at the Fermi level can be as large as hundreds of picoseconds due to the blocked spin-flip scattering[34,35]. Considering that the half-metallicity is strongly dependent on the band structure, the spin dynamics in MWSM $Co_3Sn_2S_2$ is an ideal subject to investigate the correlations between magnetization, Weyl nodes and half-metallicity.

Here, by using the femtosecond laser-induced time-resolved magneto-optical Kerr effect (MOKE) and reflectivity technique, we have revealed an ultrafast magnetization enhancement component upon exciting the ferromagnetic $Co_3Sn_2S_2$ crystal, along with spin-flipping dominated fast demagnetization and slow demagnetization due to the overall half-metallic electronic structures. We have demonstrated that this ultrafast magnetization enhancement only occurs when the temperature is below the Weyl nodes annihilation point. Our first-principle calculations show that the Weyl nodes play a crucial role in the electron relaxation and the local spin polarization around the Weyl nodes in the momentum space, which leads to the experimental observations. These results experimentally verify the proposed model accounting for the type-II Weyl nodes correlated ultrafast magnetization enhancement, which may be generally applicable to the other similar type-II magnetic Weyl semimetals.

## Results

$Co_3Sn_2S_2$ is a Shandite compound with a layered crystal structure, consisting of a kagome $Co_3Sn$ layer sandwiched between two hexagonal S layers, which are further sandwiched between two hexagonal Sn layers (Fig. 1g). The ferromagnetism of $Co_3Sn_2S_2$ arises mainly from Co, with a magnetic moment of about 0.33 $\mu_B$/Co along the c axis. The Curie temperature is ~170 K, as shown in Fig. 1h. We employed femtosecond laser-excited time-resolved MOKE and reflectivity measurements of $Co_3Sn_2S_2$ with various temperatures and pump fluences (Fig. 1i).

Figure 1j displays the time evolution of the linearly polarized laser pump-induced MOKE rotation change of $Co_3Sn_2S_2$ with different pump fluence at 10 K. An initial peak around zero delay is observed, and its

amplitude is proportional to the pump fluence (Supplementary Fig. 7a). This peak is attributed to the so-called dichroic bleaching or state blocking, which reflects the breakdown of the proportionality between the magneto-optical response and the magnetization[36,37]. Nevertheless, this non-magnetization response only lasts for hundreds of femtoseconds (the characteristic time in our measurements is <200 fs), and the subsequent magneto-optical response predominantly reflects the genuine spin dynamics (details as described in the Supplementary Information). When the pump fluence ($F_{pump}$) is as low as 0.14 mJ/$cm^2$, the spin dynamics present a sub-picosecond demagnetization followed by a magnetization recovery process, which is similar to the behavior in magnetic transition metals[36,38,39]. Surprisingly, it is found that when the pump fluence increases, two additional dynamic magnetization processes are observed. In the high pump fluence range (>1.70 mJ/$cm^2$), the magnetization dynamics are dominated by a slow demagnetization component, reflecting a half-metallic nature to block the spin-flip process and the weak spin-lattice coupling[34,35,40]. More intriguingly, a third dynamic process corresponding to an enhancement of the transient magnetization can be unambiguously recognized when the pump fluence is in a medium range (0.28–1.13 mJ/$cm^2$). The temporal traces of transient Kerr rotation changes clearly show that the net magnetization changes become positive at a timescale of tens of picoseconds. The observation of three mutually competing magnetization dynamic processes, consisting of a fast demagnetization component, a slow demagnetization component and a magnetization enhancement component, respectively, demonstrates an exotic magnetization dynamics in $Co_3Sn_2S_2$ excited by a femtosecond laser pulse.

In order to quantitatively interpret the three magnetization dynamic components, the transient MOKE rotation change is numerically fitted using the equation as follows:[41–43]

$$\left\{ \begin{array}{c} \left[ A_1\left(e^{t/\tau_{enh}} - e^{t/\tau_{r1}}\right) + A_2\left(e^{t/\tau_{fast}} - e^{t/\tau_{r2}}\right) + A_3\left(e^{t/\tau_{slow}} - e^{t/\tau_{r3}}\right)\right]\Theta(t) \\ + A_4 \Big/ \left(1 + \left(\frac{t}{\tau_4}\right)^2\right) \end{array} \right\} * G(t),$$

(2)

where $G(t)$ is the Gaussian laser pulse, * represents the convolution product, $\Theta(t)$ is the step function. $A_1$ represents the value of the enhancement component, and $\tau_{enh}$ is the characteristic magnetization enhancement time. $A_2$ and $A_3$ represent the values of the fast and slow demagnetization components, respectively, with $\tau_{fast}$ and $\tau_{slow}$ being the corresponding demagnetization times. $\tau_{r1}$, $\tau_{r2}$ and $\tau_{r3}$ are the relaxation times of each component. $A_4 \Big/ \left(1 + \left(\frac{t}{\tau_4}\right)^2\right)$ is a Lorenz pulse used to fit the initial peak, with the full width of half maximum (FWHM) of this peak being $\sqrt{2}\tau_4$. As shown in Fig. 1k, the fitted curves of each components with characteristic times $\tau_{enh} = 2$ps, $\tau_{fast} = 300$fs and $\tau_{slow} = 6$ps allow for clear distinction of the aforementioned three magnetization dynamic processes. Details of the data fitting process are included in the Supplementary Information. The overall fitted curve (red solid line in Fig. 1j) is in good agreement with the experimentally measured data (circles in Fig. 1j). The amplitudes of each component as a function of the pump fluence are shown in Supplementary Fig. 7a–d. The amplitude of $A_1$ gradually increases and reaches a saturated value when $F_{pump}$ is >1.5 mJ/$cm^2$ while the amplitude of $A_2$ is comparatively small and has no significant pump fluence dependence. The slow demagnetization component is weak when the pump fluence is low, and it becomes dominant when $F_{pump}$ is >1.0 mJ/$cm^2$, as shown in the ratios of $A_3/A_2$ and $A_3/A_1$ in Supplementary Fig. 7e, f, respectively. The summation of $A_1 + A_3$ is plotted in Supplementary Fig. 7g. There is a drastic change in behavior around the pump fluence of around 1.0 mJ/$cm^2$. The positive values below the critical pump fluence of 1.0 mJ/$cm^2$ and the negative values above it clearly indicate a transition between magnetization enhancement and slow magnetization demagnetization.

The fast demagnetization with $\tau_{fast} = 300\,fs$ can be explained by the spin-flip process (mediated by phonon[38] or magnon[44]) and the slow demagnetization with $\tau_{slow} = 6\,ps$ is attributed to the half-metallic slow demagnetization process, respectively. However, the magnetization enhancement process with a characteristic time $\tau_{enh} = 2\,ps$ cannot be interpreted by the known existing mechanisms that account for the transient enhanced MOKE response observed in other magnetic materials or heterostructures (see Supplementary Information)[29,30,45]. The question naturally arises whether this unusual ultrafast magnetic response is associated with the Weyl properties.

To investigate the underlying physical mechanism of transient magnetization enhancement, temperature dependent TR-MOKE measurements were performed. As shown in Fig. 2a, an interesting evolution of magnetization dynamics with respect to temperature is illustrated with a low pump fluence of 0.57 mJ/cm². Because of the small pump fluence, the magnetization dynamics are dominated by the fast demagnetization and magnetization enhancement components when the temperature is low. However, as the temperature increases, the magnetization enhancement component becomes weak, and this process completely disappears at a transition temperature around 110 K. When the temperature continues to increase, slow demagnetization dominates magnetization evolution. The amplitudes of components $A_1$, $A_2$ and $A_3$ as a function of temperature are shown in Fig. 2c, and a transition temperature is localized in the gray marked area. Note that the $A_2$ fast demagnetization component also decreases with

increasing temperature but completely vanishes at about 150 K. When the temperature exceeds the Curie temperature of 170 K, there is no magnetic response to ultrafast laser pulse excitation. The critical temperature around 110 K, where the magnetization enhancement component disappears, is verified with different pump fluences ranging from 0.28 – 2.26 mJ/cm², as shown in Supplementary Fig. 8 and 9. Notably, even with the lowest pump fluence of 0.28 mJ/cm² and the smallest contribution from $A_3$ slow demagnetization, the magnetization enhancement component still clearly vanishes after the transition temperature around 110 K. Additionally, pump helicity-dependent measurements are also performed, and no significant differences were observed with different temperatures and pump fluences. (See Fig. 2b and Supplementary Fig. 10).

To investigate the role of Weyl nodes in the observed ultrafast magnetization enhancement process, magnetization- (equivalent temperature $T_e$-) dependent electronic band structures were theoretically calculated. The equivalent temperature $T_e$ corresponding to each magnetic moment on the Co atoms was calibrated using the field-cooled M-T curve with a large applied field of 1 Tesla (see Supplementary Fig. 11). The band structures along high-symmetry paths are shown in Supplementary Fig. 12. As shown in Supplementary Fig. 14a, the spin-dependent density of states (DOS) at the ground state shows a spin-up dominant state at the Fermi level, which accounts for the half-metallic slow demagnetization ($A_3$) component in the transient magnetization dynamics. Although a high pump fluence is required to

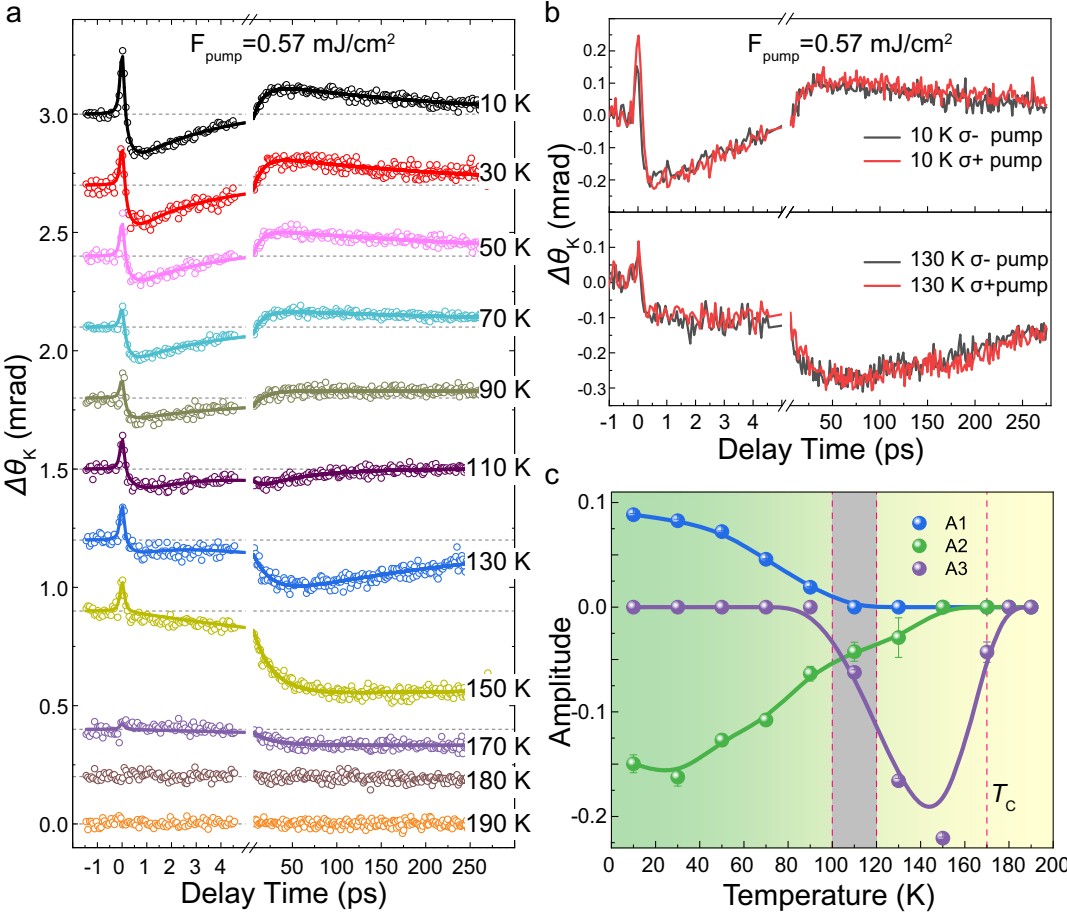

**Fig. 2 | Magnetization dynamics of Co₃Sn₂S₂ measured as a function of the temperature. a** The measured transient Kerr rotation of Co₃Sn₂S₂ at temperature varying from 10 K to 190 K, the pump fluence is 0.57mJ/cm². Circles are experimental data and the solid lines are the fitted curves. **b** The transient Kerr rotation stimulated by the left handed and right handed pump pulses with a fluence of

0.57 mJ/cm². No signification difference is observed at 10 K and 130. **c** Amplitudes of the components A1, A3, and A3 as a function of the temperature. A critical temperature around 110 K is marked in as the gray area. Curie temperature is shown by the dashed red line.

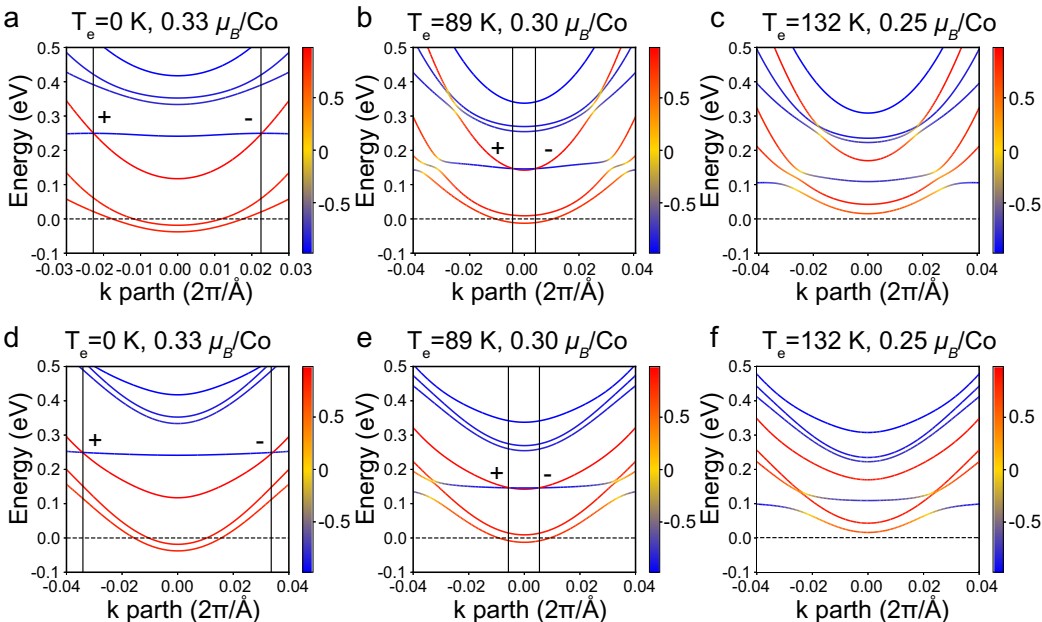

**Fig. 3 | Temperature-dependent band structures around type-II Weyl nodes by first-principle calculation.** Band structures along the momentum path passing through **a**–**c** three pairs of Weyl nodes related to $C_{3z}$ symmetry, **d**–**f** a pair of Weyl nodes related to inversion symmetry with opposite chirality (indicated by the vertical lines) and the $\Gamma$ point (zero point) at equivalent temperature $T_e = 0$, 89, and 132 K (corresponding to local magnetic moment on Co = 0.33, 0.30, and 0.25 $\mu_B$/Co). The bands with $+1/2$, 0, and $-1/2$ spin polarization are colored in red, yellow, and blue, respectively.

stimulate this half-metallic component (see Supplementary Fig. 7c), it is robust with respect to the temperature below the Curie temperature (see Supplementary Fig. 8 and 9).

We investigate the local electronic structures around the Weyl nodes as a function of $T_e$ (local magnetic moment on Co). As shown in Fig. 3 and Supplementary Fig. 13, four pairs of Weyl nodes are located around the $\Gamma$ point and the Weyl nodes indicated by the vertical lines have opposite chirality. As shown in Supplementary Fig. 13, three pairs of Weyl nodes are related by $C_{3z}$ rotation symmetry while the other one is related by inversion symmetry. Details of the Cartesian coordinates of the four pairs of Weyl nodes are presented in Supplementary Table 1. Figure 3a–c show the temperature-dependent local electronic structures of the three pairs of Weyl nodes, while Fig. 3d–f represent the one pair of Weyl nodes. It is clearly shown that all the Weyl nodes are type-II Weyl nodes with crossing spin-up and spin-down bands. The flat spin-down band and spin-up band with large dispersion are key features for the local spin filter effect via electron relaxation, as proposed in Fig. 1. Additionally, the Fermi level is dominated by the spin-up bands, forming the local half-metallicity with spin-up polarization, which accounts for the transient magnetization enhancement. Importantly, as temperature increases, the four pairs of Weyl nodes become closer to the $\Gamma$ point in the k space (Supplementary Fig. 13). These Weyl nodes annihilate between 89 K (0.30 $\mu_B$/Co) and 132 K (0.25 $\mu_B$/Co), which is in good agreement with the experimentally observed transition temperature of around 110 K for the transient magnetization enhancement.

We also investigate the other three pairs of equivalent Weyl nodes in total due to the $C_{3z}$ rotation symmetry and $C_{2y}T$ symmetry[12]. However, the Weyl nodes annihilation temperature of these Weyl nodes is found to be ~153 K, which is much higher than the observed transition temperature. Moreover, as shown in Supplementary Fig. 14, the energy of these Weyl nodes is higher (~0.5 eV at 0 K) than the abovementioned ones (~0.25 eV at 0 K), which may introduce other non-spin conserved scattering when electrons relax towards the Fermi level. This effect could be pronounced as there is a spin-down band under the Weyl nodes with topological trivial crossings. Therefore, these three pairs of Weyl nodes are not responsible for the observed magnetization

enhancement. Regarding the Weyl nodes at 60 meV above Fermi level, the local spin polarization around them remains unchanged for different temperatures, and no temperature-dependent Weyl nodes annihilation is found (see Supplementary Fig. 15).

Now, as schematically illustrated in Fig. 4, the ultrafast magnetization enhancement in Co$_3$Sn$_2$S$_2$ correlated with type-II Weyl nodes stimulated by femtosecond laser pulses can be interpreted through the following process. When the temperature is lower than the transition temperature ~110 K, pump pulse excitation with a wavelength of 800 nm generates spin-polarized or unpolarized electrons through circularly polarized or linearly polarized light, which are far above the Fermi level and much higher than the Weyl nodes in the momentum space. Subsequently, the excited state electrons relax into lower states near the upper Weyl cones via intraband transitions, which is mainly attributed to electron-electron scattering that causes dephasing of the spin polarization of the electrons. This also explains the similar experimental observations using different polarized light pulses. When the electrons relax into the Weyl cones, they are topologically protected with spin-momentum locking (in x-y plane). However, the lower Weyl cones are composed of spin-up bands, causing local spin polarization states underneath the Weyl nodes. Additionally, as the Weyl cones are tilted (type-II Weyl semimetals), the energy dispersion at the Weyl nodes differs between the spin-up and spin-down bands, resulting in different electron-electron scattering rates for the two spin channels when the electrons pass through the Weyl nodes, as illustrated in Fig. 1. Consequently, spin-up electrons transit the Weyl nodes much faster than the spin-down electrons. The lower bands under the Weyl nodes are therefore spin (up) polarized, forming a spin-up polarized pocket (see Supplementary Fig. 17). Furthermore, states at the Fermi level under the Weyl cones are spin-up polarized. Due to the Stoner model, the elevated density of states with up spins near the Fermi level enhances the magnetization ultimately. In this proposed picture, the Weyl node acts as a spin filter during the electron relaxation in momentum space, accounting for the observed ultrafast magnetization enhancement. When the temperature is higher than ~110 K, the transition from other parabolic bands may superimpose over the response from the bands

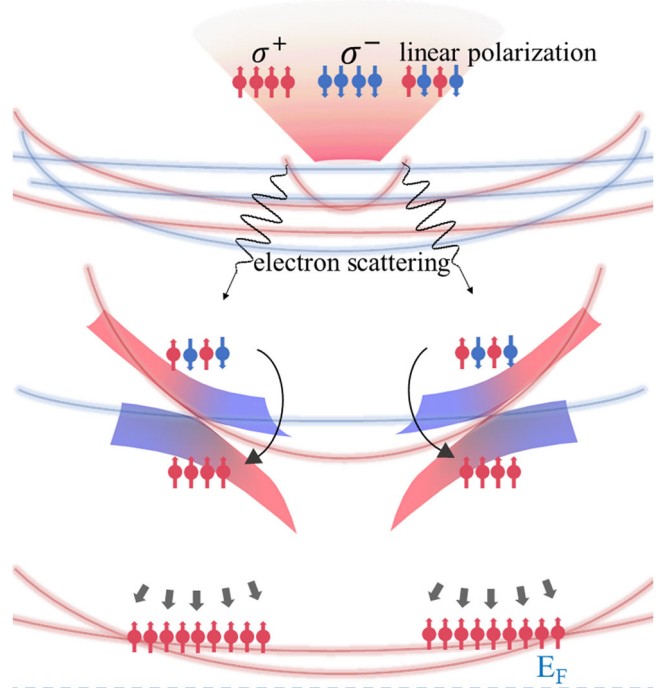

**Fig. 4 | Weyl nodes correlated electron relaxation model corresponding to the ultrafast magnetization enhancement.** Spin polarized or unpolarized electrons are generated by circularly or linearly light, which are far above the Fermi level and much higher than the Weyl nodes. Up spin and down spin are colored in red and blue, respectively. The excited state electrons relax into lower states near the upper Weyl cones via the intraband transitions. The electrons are unpolarized due to the electron-electron scattering. The lower Weyl cones are composed by the spin-up bands, causing the local spin-polarization states underneath the Weyl nodes, forming a spin-up polarized pocket. The spin-flipping rate for the spin-up and spin-down electrons are different due to the energy dispersion at type-II Weyl nodes. The spin-up electrons transit the Weyl nodes in a much faster relaxation time compared to the spin-down electrons. Also, the states at the Fermi level under the Weyl cones are spin up polarized. Due to the Stoner model, the elevated density of states with up spins near the Fermi level enhance the magnetization ultimately. The type-II Weyl node acts as a *spin filter* during the electron relaxation in the momentum space.

near Weyl nodes due to the magnetization-induced band shift[8]. Thus, the slow demagnetization process ($A_3$) becomes dominant due to the half-metallic nature. We note that the relaxation time ($\tau_{r3}$) of the slow demagnetization process ($A_3$) significantly increases with temperature, which is attributed to the band gap at the Weyl nodes. This observation confirms the crucial role of the Weyl nodes in the ultrafast time-resolved magnetization dynamics in $Co_3Sn_2S_2$.

We now discuss the magnetic ordering time of the observed magnetization enhancement. The magnetization enhancement characteristic time of 1–2 ps corresponds to an exchange coupling energy of 2–4 meV[46], which is strikingly small compared to that of typical 3d itinerant magnet. This value is in good agreement with the calculated nearest-neighbor exchange coupling $J_1 = 2.37$ meV[47] in $Co_3Sn_2S_2$. We note that due to the kagome lattice of cobalt ions in $Co_3Sn_2S_2$, the exchange couplings are very complicated. It has recently been reported that an unusual third-neighbor "across-hexagon" exchange coupling $J_d$ dominates the ferromagnetic ground state[10,48,49]. Thus, further investigations are needed to clarify which exchange coupling(s) dominate the magnetic ordering in our observations here.

Besides the exchange energy, the time for collective magnetic ordering to build up depends on how fast electronic correlations spread through the crystal. For example, it has been demonstrated that although the exchange energy of EuGdO is around 0.1 eV, corresponding to a timescale of ~40 fs, a magnetic ordering time of ~3 ps was

observed[46]. The propagation velocity of the magnetic excitation is reduced by the ratio of the effective masses of the 5d6s conduction electrons and the heavy holes in the 4 f band. Here, in $Co_3Sn_2S_2$, the exchange coupling $J_d$ occurs via Co-Sn-Co bond across the Co hexagon, and the conduction electron group velocity may also depend on the effective masses of the bands of Co and Sn. This could significantly enlarge the magnetic build-up time towards picosecond timescale.

## Discussion

In summary, we have investigated the ultrafast spin dynamics in the half-metallic type-II Weyl semimetal $Co_3Sn_2S_2$ with a ferromagnetic kagome lattice. Upon femtosecond laser pulse excitation, we observed an exotic transient magnetization behavior consisting of three components, following the initial non-thermal non-magnetic peak. In addition to the fast demagnetization dominated by spin-flipping and the slow demagnetization due to the overall half-metallic electronic structures, an unexpected magnetization enhancement is observed on a picosecond timescale. Temperature and pump fluence-dependent measurements show that this process occurs only when the temperature is lower than the critical Weyl property transition temperature (-110 K), demonstrating that this ultrafast magnetization enhancement is associated with the Weyl nodes. By analyzing the theoretical calculated electronic structures, we propose that the local spin polarization around the Weyl nodes in momentum space plays a key role. The type-II Weyl nodes act as a spin filter when the excited electrons pass through, resulting in the enhancement of the DOS with up spins near the Fermi level. The measured transient magnetization is therefore enhanced, lasting for hundreds of picoseconds. Our results not only reveal the unique correlated roles of the half-metallicity and Weyl properties in $Co_3Sn_2S_2$ in femtosecond laser-induced spin dynamics but also open avenues for studying the fascinating ultrafast phenomena in magnetic Weyl semimetal. We believe our physics model can be generally applied to other potential magnetic type-II Weyl semimetal with similar electronic structures.

## Methods

### Single-crystal growths

$Co_3Sn_2S_2$ single crystals were grown using Sn-flux method with Co:Sn:S molar ratio of 4:43:3. Stoichiometric amounts of Co, Sn and S powders were mixed and placed in an alumina crucible, which was then sealed in a quartz ampoule under vacuum. The sealed quartz ampoule was heated to keep at 1323 K and held at that temperature for 6 h, then slowly cooled down to 973 K with a rate of 5 K/h. Subsequently, the crucible was taken out from the furnace and decanted with a centrifuge to separate $Co_3Sn_2S_2$ crystals from Sn flux. After the centrifugation process, most of the flux contamination was removed from the surface of crystals. X-ray diffraction (XRD) and energy-dispersive X-ray spectroscopy (EDS) results (Supplementary Fig. 1) show the high-quality of our sample.

### Time-resolved MOKE and reflectivity measurements

The time-resolved magneto-optic Kerr effect (TR-MOKE) and time-resolved reflectivity (TR-R) measurements are schematically shown in Fig. 1c. The mode-locked amplified Ti: Sapphire laser was used to emit pulses with a wavelength of 800 nm and a pulse duration of 50 fs at a repetition frequency of 1000 Hz. The linear or circularly polarized pump beam with 800 nm wavelength is modulated by an optical chopper at a frequency of 333 Hz and this frequency is used as a reference for the lock-in amplifiers. After the pump pulse excitation, the delayed probe beam with 400 nm wavelength, which is generated through a beta barium borate (BBO) in order to be distinguished with the pump beam so that the signals could not be obscured, incident onto the sample surface to measure the transient MOKE and reflectivity change. The pump beam is normally incident onto the sample while the probe beam is a slight angle away from the pump beam. The time delay[46] between the pump and probe is precisely controlled by an

optical delay line. The laser spot sizes of pump beam and probe beam are 300μm and 100μm, respectively. The reflected probe beam is split into two half. One portion is used to measure the transient reflectivity using a single potodiode detector. The other portion of the probe is polarization-dependently split into two beam using the Wollaston prism and by applying a half wave-plate, an optical bridge detector is used to monitor the very small Kerr rotation signal. An external magnetic field of $\pm 5000$ Oe is applied along the easy axis (c axis in Fig. 1a) of the sample which is high enough to saturate the crystal even at the lowest temperature. The sample is loaded in a cryostat with an optical window for the temperature dependent experiments. The transient Kerr rotation change $\Delta\theta_K(\tau)$ is defined in the following as the asymmetric part, changing with the field direction $\Delta\theta_K(\tau) = (\Delta\theta_K(\tau, M) - \Delta\theta_K(\tau, -M))/2$. The time-resolved reflectivity (TR-R) measured in opposite external magnetic fields show no difference at a certain pump fluence so that the average of them has been used to represent the reflectivity dynamics. (see Supplementary Fig. 2)

### Magnetization and transport measurements

The bulk magnetic measurements were performed in a Quantum Design SQUID VSM magnetometer with the magnetic field of 50 Oe and temperature ranging between 2 and 300 K. The electrical transport measurements were performed in a Quantum Design PPMS system, in which the lowest temperature and the highest magnetic field are 1.9 K and 9 T, respectively.

### Theoretical calculation

We calculate the electronic structure of $Co_3Sn_2S_2$ using the Vienna ab initio simulation package (VASP)[50] with the generalized gradient approximation of Perdew-Burke-Ernzerhof exchange-correlation potential[51]. The self-consistent calculation was carried out on an $11 \times 11 \times 11$ k-mesh with the energy cutoff of 400 eV. We fixed the magnetic moment to simulate the varying temperatures from 0–175 K[12]. The maximally-localized Wannier functions[52–54] were generated using d orbital of Co, p orbital of Sn and p orbital of S by the WANNIER90 package[55].

## Data availability

The data that support the findings of this work are available within the paper and its Supplementary Information. Additional data are available from the corresponding authors upon request.

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

## Acknowledgements

This work is supported by the National Key Research and Development Program of China (Grant Nos. 2021YFB3601600, 2018YFA0305700 and No. 2022YFA1403800), the National Natural Science Foundation of China (Grant Nos. 12104216, 12241403, 92165201, 61427812, 11925408, 11921004 and 12188101), the Natural Science Foundation of Jiangsu Province of China (Nos. BK20200307, BK20192006, and BK20180056), the Chinese Academy of Sciences (Grant No. XDB33000000), the Informatization Plan of the Chinese Academy of Sciences (Grant No. CAS WX2021SF-0102), the Condensed Matter Physics Data Center, CAS the Innovation Program for Quantum Science and Technology (Grant No. 2021ZD0302800), the CAS Project for Young Scientists in Basic Research (Grant No. YSBR-046), and the China Postdoctoral Science Foundation (Grant Nos. 2021M703104 and 2022T150629).

## Author contributions

X.L. and Y.X. conceived the project. Z.L., H.Z., L.L., and C.Z. prepared Co3Sn2S2 crystal and performed SQUID and magnetic transport measurements. X.L. and G.L. performed the TR-MOKE and TR-R experiments with the help from X.R., Y.G., Y.Y., J.W., and Y.L. H.P., T.Z., and H.W. performed the density functional theory calculations. X.L., H.P., H.W., Z.L., J.W., L.H., R.Z., and Y.X. performed the data analysis. X.L., H.P., Z.L., and Y.X. wrote the paper with contributions from all authors. All authors discussed the results, interpretation and conclusion.

## Competing interests

The authors declare no competing interests.

## Additional information

[1]School of Integrated Circuits, Nanjing University, Suzhou 215163, China. [2]State Key Laboratory of Spintronics Devices and Technologies, Nanjing University, Suzhou 215163, China. [3]Jiangsu Provincial Key Laboratory of Advanced Photonic and Electronic Materials, School of Electronic Science and Engineering, Nanjing University, Nanjing 210093, China. [4]International Center for Quantum Design of Functional Materials (ICQD), Hefei National Laboratory for Physical Sciences at the Microscale, and Synergetic Innovation Center of Quantum Information and Quantum Physics, University of Science and Technology of China, Hefei, Anhui 230026, China. [5]CAS Key Laboratory of Strongly-Coupled Quantum Matter Physics, and Department of Physics, University of Science and Technology of China, Hefei, Anhui 230026, China. [6]Beijing National Research Center for Condensed Matter Physics, Institute of Physics, Chinese Academy of Sciences, Beijing 100190, China. [7]School of Physical Science, University of Chinese Academy of Sciences, Beijing 100049, China. [8]Songshan Lake Materials Laboratory, Dongguan, Guangdong 523808, China. [9]Department of Chemistry, University of Pennsylvania, Philadelphia, PA 19104-6323, USA. [10]School of Integrated Circuits, Guangdong University of Technology, Guangzhou 510006, China. [11]York-Nanjing International Joint Center in Spintronics, School of Physics, Engineering and Technology, University of York, York YO10 5DD, UK. [12]These authors contributed equally: Xianyang Lu, Zhiyong Lin, and Hanqi Pi. ✉e-mail: jing.wu@gdut.edu.cn; hmweng@iphy.ac.cn; cgzeng@ustc.edu.cn; ybxu@nju.edu.cn

