## [Peer Review File · Nature Communications]

REVIEWER COMMENTS

Reviewer #1 (Remarks to the Author):

The present paper reports intriguing magneto-response of the Co₃Sn₂S₂ crystal, the magnetic type-II Weyl semimetal. Depending on the pump beam flux or the temperature, temporal variations of the Kerr-rotational angle indicate a combination of the enhanced magnetization and the slow demagnetization processes. The phenomena were described by electronic band structure that is temperature dependent. A unique spin-polarized band of the type-II Weyl node has acted as a dynamic spin filter. My comments are following.

1) Time-resolved data of the magneto-optical Kerr rotations are interesting to find rich dynamics of magnetization. The data analysis was made with a simple phenomenological equation, (2) in the paper, that deconvolutes the dynamic events under the exponential approximation. Thus, the main discussion quite depends on interpretations by the authors. A key comparison between theory and experiment was the temperature dependence. I find the band model, based on the magnetic type-II Weyl semimetal, is plausible. However, in the manuscript, there are several types of the temperatures that can be dealt. The first case of the temperature is given in Fig.1 and Fig.2 that likely monitors the sample temperature at the measurement stage. The second case is adopted in Fig.3 for the calculation and the temperature was determined by the M-T curve in Fig.S10. The third case, not mentioned in the paper, is the transient temperature that is realized in the ultrafast timescale after the optical pumping. During the fs-ps timescale, the energy transfer was made into phonons and thus the sample temperature is most likely different from the first case. I infer from the paper that the authors used the M-T curve in the second case to consider the transient temperature of the third case. However, since it was not explicitly described in the paper, it remains uncertain how to compare the equilibrium and the non-equilibrium temperatures. The paper becomes better when the authors elaborate treatment of the temperature in the discussion.

2) One of the exciting data is the transition between the magnetization and demagnetization that can be found at the pump flux of 1.13~1.7 mJ/cm². The change directly links to the flux dependence of A₁ and A₃ that is summarized in Fig.S6. I think the A₃/A₁ ratios given in Fig.S6(f) should have negative values. On the other hand, according to Eq.(2) in the paper, the summation, A₁ + A₃, should give a drastic change at the pump flux of 1.13~1.7 mJ/cm².

3) In Fig. 3(c) and (f), vertical lines of the chirality appeared differently. It should be mentioned in the paper.

Reviewer #2 (Remarks to the Author):

Magnetization dynamics is a very interesting topic with many questions still open from both a theoretical and an experimental view point.

In this work the authors explore the magnetization dynamics of a type-II MWSM $\text{Co}_3\text{Sn}_2\text{S}_2$ crystal. By using MOKE they reveal a novel ultrafast magnetization enhancement effect, along with a spin flipping dominated fast demagnetization and a slow magnetization dynamics due to half metallicity (spin flips are forbidden in this regime). They found that the magnetization enhancement process is caused by electronic excitations that relax into up spin polarized topologically protected Weyl cones.

Overall the paper is very interesting. However, as a general comment I would appreciate a bit more rigor and at least attempts to corroborate some of the claims made throughout the paper with some, more theoretical explanation.

As it is the paper gives me the impression that it is more descriptive. It is true that the authors try to explain their findings, but overall I found this part of the work not sufficient for such a high standard journal.

I would recommend publication after the authors try to clarify at least these three crucial points:

1) The fast demagnetization phase. The authors say that it lasts approximately 300 fs. This looks reasonable to me however, this time scale is quite long if you claim that fast demagnetization is entirely due to electron-electron relaxation. I could be wrong, but this time scale is more typical of electron-phonon scattering mechanisms. Later you explain the slow dynamics in terms of weak spin-phonon interactions. Please, I would be curious to know, and I believe the paper will considerably improve if you give a more detailed explanation about the fast processes (I understand it is not central for the paper, but given that you talked about it I would appreciate you provide more details).

2) Eq. (1) at line 117. I think it is useless in this form for what you want to prove. First of all it is for a free electron model. Is it justified here? Below you say that τ_{ee} differs between spin up and spin down components because of overtilted Weyl cones. How does that relaxation coefficient change?

3) For the magnetization enhancement part. You provide an explanation, but I do not see exactly how the change in population in the Weyl nodes affect the magnetization enhancement and how this changes with temperature. This part is really obscure to me.

In addition, please look for typos.

At line 312 I do not know what it means electrons transmit the Weyl nodes.

At line 300 circular or linear light.

Overall I found the paper interesting and I would be happy to recommend for publication after these problems have been taken in consideration

Reviewer #3 (Remarks to the Author):

Lu et al. present the results of time resolved optical measurements performed upon $\text{Co}_3\text{Sn}_2\text{S}_2$ crystals. They record both the time resolved reflectivity and Kerr rotation, and the most significant finding is that for a certain range of pump fluence and sample temperature, an enhancement of the magnetization may be observed. The electronic band structure is calculated with the VASP package, to confirm that $\text{Co}_3\text{Sn}_2\text{S}_2$ is a magnetic type-II Weyl semimetal (MWSM). The transient enhancement of the magnetization is attributed to spin-filtering at some of the observed Weyl points. This interpretation is supported by the observed temperature dependence of the enhancement, which is only observed at temperatures below 110K in the experiments, before the annihilation of the Weyl points at 0.25 eV above the Fermi level that occurs in the simulations at this temperature.

The results may extend understanding of the MWSM and offer a new mechanism for the manipulation of spin dynamics in quasi-2D van der Waals bonded materials, which is likely to be of interest to researchers studying both 2D materials and magnetism and spintronics. However, the interpretation makes many assumptions and seems rather speculative. The following issues also need to be addressed.

1. Few details are given of the $\text{Co}_3\text{Sn}_2\text{S}_2$ crystals. What is their size and shape, and what characterisation has been performed of structure and composition?

2. The explanation of the time resolved optical response depends upon electrons being excited above the Fermi level by a pump beam of 800 nm (photon energy 1.55 eV). Some indication should be given of the electronic transitions induced by the pump beam. While it may be difficult to tune the wavelength of the beam, it should be possible to refer to published optical spectra and the calculated band structure.
3. More justification should be given for the claimed bottleneck effect. The opening of a gap at the Weyl point means that finite momentum is required for scattering across the gap. Is this the source of the bottleneck and how does the scattering rate scale with the size of the gap if electron-electron scattering is the dominant scattering mechanism?
4. The authors should be more explicit about the nature of the half-metallic slow demagnetization mechanism. I assume they mean that the minority density of states at the Fermi level increases, allowing increased relaxation of majority spins. Please confirm or explain otherwise.
5. Time resolved optical data is frequently discussed in the context of a two or three temperature model. Is this a useful approach in the present case?
6. The reliability of the fitting of the magneto-optical data is unclear. Fits of data acquired at a temperature of 10 K are presented in which τ_{enh} , τ_{fast} , and τ_{slow} are fixed, but the associated recovery relaxation times τ_{r1} , τ_{r2} , and τ_{r3} are allowed to vary. Table S2 shows that the fitted recovery times exhibit significant variation as the pump fluence is varied and may compensate for variations in τ_{enh} , τ_{fast} , and τ_{slow} . Therefore the assumption of fixed τ_{enh} , τ_{fast} , and τ_{slow} requires further justification.
7. The temperatures used to label the VASP calculations are confusing at present. My impression is that the calculations are performed at zero temperature, but the authors are able to fix the moment on the Co ions. Therefore, they use the measured SQUID data in Figure S10 to determine the moment corresponding to each temperature, and then assume that this will produce the correct band structure for that temperature. Please either confirm or explain otherwise. And what is the justification for such an approach?
8. I think by “electrons transmit the Weyl nodes” that authors actually mean “electrons transit the Weyl nodes”, but please confirm.
9. In figures S7 and S8 the captions refer to fitted curves but I don't see any curves in the plots.
10. The specular optical Kerr effect (SOKE) vanishes when pump and probe polarization are the same, or if the pump has circular polarization, but is maximised when the planes of polarization of pump and probe lie 45° apart. The authors talk about SOKE in the Supplementary Material, so did they also measure the SOKE with planes of polarization of pump and probe lie 45° apart? To fully understand the zero delay peak it is also necessary to measure the transient Kerr ellipticity. Were such measurements made?

Response to Reviewers' Comments

Response to the comments of Reviewer #1

Introductory comment:

The present paper reports intriguing magneto-response of the $\text{Co}_3\text{Sn}_2\text{S}_2$ crystal, the magnetic type-II Weyl semimetal. Depending on the pump beam flux or the temperature, temporal variations of the Kerr-rotational angle indicate a combination of the enhanced magnetization and the slow demagnetization processes. The phenomena were described by electronic band structure that is temperature dependent. A unique spin-polarized band of the type-II Weyl node has acted as a dynamic spin filter. My comments are following.

Response to comment:

We are grateful for the reviewer's careful review and his/her positive comments. We have revised our manuscript to address the reviewer's comments and suggestions.

Comment #1:

Time-resolved data of the magneto-optical Kerr rotations are interesting to find rich dynamics of magnetization. The data analysis was made with a simple phenomenological equation, (2) in the paper, that deconvolutes the dynamic events under the exponential approximation. Thus, the main discussion quite depends on interpretations by the authors. A key comparison between theory and

experiment was the temperature dependence. I find the band model, based on the magnetic type-II Weyl semimetal, is plausible. However, in the manuscript, there are several types of the temperatures that can be dealt. The first case of the temperature is given in Fig.1 and Fig.2 that likely monitors the sample temperature at the measurement stage. The second case is adopted in Fig.3 for the calculation and the temperature was determined by the M-T curve in Fig.S10. The third case, not mentioned in the paper, is the transient temperature that is realized in the ultrafast timescale after the optical pumping. During the fs-ps timescale, the energy transfer was made into phonons and thus the sample temperature is most likely different from the first case. I infer from the paper that the authors used the M-T curve in the second case to consider the transient temperature of the third case. However, since it was not explicitly described in the paper, it remains uncertain how to compare the equilibrium and the non-equilibrium temperatures. The paper becomes better when the authors elaborate treatment of the temperature in the discussion.

Response to comment #1:

We would like to thank the reviewer for raising this important issue. In the manuscript, we used two types of temperatures. The first one, as pointed out by the reviewer, is the sample temperature at the measurement stage, as shown in Fig. 1 and Fig.2. The second one is the equivalent temperature used in the first-principle calculations. As described in the methods section, the band structures were calculated with varied fixed magnetic moment. The equivalent temperature was then calibrated using the M-T curve. This method is helpful in understanding the temperature (magnetic moment) – dependent band structures, similar to the approach used in Phys. Rev. Lett. 124, 077403 (2020) and Phys. Rev. B 104, L100301 (2021).

Regarding the transient temperature, we did not consider this non-equilibrium temperature in our

analysis for the following reasons. The key observations of the magnetization enhancement and the temperature-dependent transition of the magnetization dynamic processes are shown over a wide range of pump fluence (Fig. 1(j), Fig. 2(a), Fig. S8 and Fig. S9). Only when the pump fluence is high enough, such as 2.26 mJ/cm^2 , is the magnetization enhancement not clearly evident. Therefore, it is only in this high pump fluence case that the transient elevated non-equilibrium temperature might play a significant role. However, this is not a concern for us because the main temperature-dependent analysis in our manuscript was conducted using a very low pump fluence of 0.57 mJ/cm^2 . For this pump fluence, the magnetization enhancement process and the transition of the magnetization dynamic processes are clearly shown to be temperature (sample) dependent. Moreover, the transition temperatures are consistently around 110 K for low pump fluence, as shown in Fig. S8. Consequently, we conclude that the base sample temperature determines the magnetization dynamics for low pump fluence.

Furthermore, as calculated in Phys. Rev. B 78, 174422 (2008) and Appl. Phys. Lett. 109, 042401 (2016), for a pump fluence of 2 mJ/cm^2 , the peak electron temperature is about 1000 K, and the lattice temperature is approximately to be 389 K for quasi-equilibrium state within 10 ps for metallic CoFeB (experiments conducted at room temperature). In our measurements, we believe that the transient elevated non-equilibrium temperature cannot be a significant factor when analyzing results obtained with a low pump fluence of 0.57 mJ/cm^2 .

Revision in response to comment #1:

The sentences on page 10 “To investigate the role of Weyl nodes in the observed ultrafast magnetization enhancement process, magnetization- (temperature-) dependent electronic band structures were theoretically calculated. The equivalent temperature corresponding to each

magnetic moment on the Co atoms was calibrated using the field-cooled M-T curve with a large applied field of 1 Tesla (see Fig. S10).” has been replaced by “ To investigate the role of Weyl nodes in the observed ultrafast magnetization enhancement process, magnetization- (equivalent temperature T_e -) dependent electronic band structures were theoretically calculated. The equivalent temperature T_e corresponding to each magnetic moment on the Co atoms was calibrated using the field-cooled M-T curve with a large applied field of 1 Tesla (see Fig. S11).”

The sentence on page 11 “We investigate the local electronic structures around the Weyl nodes as a function of temperature (local magnetic moment on Co)” has been replaced by “We investigate the local electronic structures around the Weyl nodes as a function of T_e (local magnetic moment on Co)”

Figure 3 has been replaced by

Figure 3 | Temperature-dependent band structures around type-II Weyl nodes by first-principle calculation. Band structures along the momentum path passing through **a-c**, three pairs of Weyl nodes related to C_{3z} symmetry, **d-f**, a pair of Weyl nodes related to inversion symmetry

with opposite chirality (indicated by the vertical lines) and the Γ point (zero point) at equivalent temperature $T_e = 0, 89, \text{ and } 132 \text{ K}$ (corresponding to local magnetic moment on Co = 0.33, 0.30, and $0.25 \mu_B/\text{Co}$). The bands with $+1/2, 0, \text{ and } -1/2$ spin polarization are colored in red, yellow, and blue, respectively.

Comment #2:

One of the exciting data is the transition between the magnetization and demagnetization that can be found at the pump flux of $1.13\sim 1.7 \text{ mJ/cm}^2$. The change directly links to the flux dependence of $A1$ and $A3$ that is summarized in Fig.S6. I think the $A3/A1$ ratios given in Fig.S6(f) should have negative values. On the other hand, according to Eq.(2) in the paper, the summation, $A1 + A3$, should give a drastic change at the pump flux of $1.13\sim 1.7 \text{ mJ/cm}^2$.

Response to comment #2:

We appreciate the reviewer for pointing out this mistake and offering the excellent suggestion. The $A3/A1$ ratios have been corrected in the revised Supplementary Information. The summation of $A1+A3$ is plotted in Fig. S7 (g). Indeed, there is a drastic change in behavior around the pump fluence of around 1.0 mJ/cm^2 . Furthermore, the positive values below the critical pump fluence of 1.0 mJ/cm^2 and the negative values above it clearly indicate a transition between magnetization enhancement and slow magnetization demagnetization.

Revision in response to comment #2:

The paragraph “There is a drastic change in behavior around the pump fluence of around 1.0 mJ/cm^2 . The positive values below the critical pump fluence of 1.0 mJ/cm^2 and the negative values

above it clearly indicate a transition between magnetization enhancement and slow magnetization demagnetization.” has been added in page 9.

Fig. S7 in the Supplementary Information is replaced by

Figure S7| Fitted amplitudes of the magnetization dynamic components. a-d, Amplitudes of the components A1, A2, A3 and A4 as a function of pump fluence at 10 K, respectively. **e-f,** The ratios of A3/A2 and A3/A1 as a function of pump fluence at 10 K, respectively. **g, A1+A3 as a function of pump fluence at 10 K.**

Comment #3:

In Fig. 3(c) and (f), vertical lines of the chirality appeared differently. It should be mentioned in the paper.

Response to comment #3:

We thank the referee's useful suggestion. This is indicated in the revised manuscript. Also, the vertical lines in Fig. 3 (f) have been removed where no Weyl nodes exist.

Revision in response to comment #3:

The sentence on Page 11“**Four pairs of Weyl nodes are located around the Γ point**” has been replaced by “**As shown in Fig. 3 and Fig. S13, four pairs of Weyl nodes are located around the Γ point and the Weyl nodes indicated by the vertical lines have opposite chirality.**”

Fig 3 has been replaced as shown in revision in response to Comment #1 of Reviewer #1

Response to the comments of Reviewer #2

Introductory comment:

Magnetization dynamics is a very interesting topic with many questions still open from both a theoretical and an experimental view point.

In this work the authors explore the magnetization dynamics of a type-II MWSM $\text{Co}_3\text{Sn}_2\text{S}_2$ crystal. By using MOKE they reveal a novel ultrafast magnetization enhancement effect, along with a spin flipping dominated fast demagnetization and a slow magnetization dynamics due to half metallicity (spin flips are forbidden in this regime). They found that the magnetization enhancement process is caused by electronic excitations that relax into up spin polarized topologically protected Weyl cones.

Overall the paper is very interesting. However, as a general comment I would appreciate a bit more rigor and at least attempts to corroborate some of the claims made throughout the paper with some, more theoretical explanation.

As it is the paper gives me the impression that it is more descriptive. It is true that the authors try to explain their findings, but overall I found this part of the work not sufficient for such a high standard journal.

I would recommend publication after the authors try to clarify at least these three crucial points:

Response to comment:

We are grateful for the reviewer's careful review and his/her positive comments. We have revised

our manuscript to address the reviewer's comments and suggestions.

Comment #1:

The fast demagnetization phase. The authors say that it lasts approximately 300 fs. This looks reasonable to me however, this time scale is quite long if you claim that fast demagnetization is entirely due to electron-electron relaxation. I could be wrong, but this time scale is more typical of electron-phonon scattering mechanisms. Later you explain the slow dynamics in terms of weak spin-phonon interactions. Please, I would be curious to know, and I believe the paper will considerably improve if you give a more detailed explanation about the fast processes (I understand it is not central for the paper, but given that you talked about it I would appreciate you provide more details).

Response to comment #1:

We would like to thank the reviewer for providing us with an opportunity to clarify this important issue. We agree with the reviewer that the fast demagnetization cannot be attributed to electron-electron relaxation. In fact, when discussing this fast demagnetization we attribute it solely to the spin-flip process and not electron-electron scattering. The reasons for this are as follows.

The underlying physical mechanisms of ultrafast demagnetization (on a time scale of ~ 100 fs) in magnetic materials are highly complex. It has been demonstrated that even in simple transition metals such as Co, Fe, and Ni, the mechanisms differ [Phys. Rev. Lett. 119, 107203 (2017)]. Several models have been proposed. The first one involves Elliott-Yafet spin-flip scattering accompanied by the emission of a phonon [Nat. Mater. 9, 259 (2010)]. In this model, an electron

flips its spin upon emission or absorption of a phonon, described by a probability a_{sf} . Another model, magnon-mediated Elliott-Yafet spin-flip scattering, effectively explains the ultrafast demagnetization in thin Fe films [Phys. Rev. B 78, 174422 (2008)]. This model is similar to the first spin-flip model, but the medium involved in the spin-flipping process is different. The third model is superdiffusive spin transport [Phys. Rev. Lett. 105, 027203 (2010)], which involves spin-dependent transport of laser-excited electrons contributing to the ultrafast demagnetization process. However, this contribution is usually significant in magnet/non-magnet heterostructures [Nat. Commun. 3, 1037 (2012)].

Therefore, in our results, we attribute the observed fast demagnetization process in $\text{Co}_3\text{Sn}_2\text{S}_2$ to the spin-flip process. However, whether magnons or the phonons are involved in the spin-flipping process remains an open question, and a time-resolved XMCD investigation may provide more information in subsequent studies.

Revision in response to comment #1:

The sentences “At first glance, the fast demagnetization with $\tau_{fast} = 300$ fs and the slow demagnetization with $\tau_{slow} = 6$ ps can be explained by the spin flip process and the half-metallic slow demagnetization process, respectively.” has been replaced by “The fast demagnetization with $\tau_{fast} = 300$ fs can be explained by the spin-flip process (mediated by phonon³⁶ or magnon⁴²) and the slow demagnetization with $\tau_{slow} = 6$ ps is attributed to the half-metallic slow demagnetization process, respectively.” on Page 9 in the maintext.

Reference 42 is added.

Comment #2:

Eq. (1) at line 117. I think it is useless in this form for what you want to prove. First of all it is for a free electron model. Is it justified here ? below you say that τ_{ee} differs between spin up and spin down components because of overtilted Weyl cones. How does that relaxation coefficient change ?

Response to comment #2:

We thank reviewer for the valuable feedback.

We did not provide a justification for the free electron model, as the linear energy dispersion of the Weyl nodes in $\text{Co}_3\text{Sn}_2\text{S}_2$ is only observed in the region very close to the gapless points. Instead, we focused on the electrons occupying energy states away from the Weyl nodes, which can be approximately described by quadratic dispersion. Specifically, in the upper part of the Weyl cone, spin-up electrons with a larger group velocity would occupy higher energy states compared to spin-down electrons when they are equidistant from the Weyl points in reciprocal space. Consequently, the spin-up electrons exhibit a higher relaxation rate for transitioning through the Weyl nodes, as indicated by Eq. (1).

Comment #3:

For the magnetization enhancement part. You provide an explanation, but I do not see exactly how the change in population in the Weyl nodes affect the magnetization enhancement and how this changes with temperature. This part is really obscure to me.

Response to comment #3:

We would like to appreciate the reviewer for raising this issue. The magnetization enhancement process is experimentally observed to be temperature dependent and this temperature dependence is demonstrated to be consistent with the equivalent-temperature (magnetization) dependence of the Weyl nodes calculated. This observation indicates that the Weyl nodes play a key role in the magnetization enhancement process. As shown in Fig. R1 (a), the spin-dependent DOS of the ferromagnetic ground state is shown (derived from the calculated DOS in Phys. Rev. B 88, 144404 (2013), Phys. Rev. B 79, 205116 (2009), Phys. Rev. B 102, 104434 (2020)). The DOS near the Fermi level is dominated by the Co 3d bands and a large spin polarization is shown at Fermi level with a low total DOS. In our manuscript, due to the dynamic spin filter effect of the Weyl nodes as we proposed, only the spin-up electrons accumulate underneath the Weyl nodes after the electron relaxation upon pump laser excitation. In this case, a transient occupation of the spin-up states above the Fermi level is induced. Now at the transient equivalent Fermi level, the spin polarization as well as the total DOS is enhanced. Also, the total spin polarization across the Co 3d bands is enhanced. Therefore, the magnetization is enhanced. This Fermi-surface tunned magnetization enhancement is kind of similar to the observation in Phys. Rev. Lett. 125, 267205 (2020) and Nature 563, 94 (2018) while the underlying mechanism is different. We note that the strong Berry curvature may be an important factor and the exact role of Berry curvature during the dynamics worth further investigation.

Figure R1 a, Spin-dependent DOS of $\text{Co}_3\text{Sn}_2\text{S}_2$ in ferromagnetic ground state. (derived from the calculated DOS in Phys. Rev. B 88, 144404 (2013), Phys. Rev. B 79, 205116 (2009), Phys. Rev. B 102, 104434 (2020)) **b**, The transient occupation of the spin-up states above the Fermi level due to the dynamic spin filter effect of Weyl nodes.

Comment #4:

In addition, please look for typos

.At line 312 I do not know what it means electrons transmit the Weyl nodes.

At line 300 circular or linear light.

Response to comment #4:

We thank the referee for careful review and apologize for the typo and language errors. The language has been carefully checked throughout our revised manuscript.

Comment #5:

Overall I found the paper interesting and I would be happy to recommend for publication after these problems have been taken in consideration

Response to comment #5:

We would like to express our gratitude to the reviewer once again for the positive comments.

Response to the comments of Reviewer #3

Introductory comment:

Lu et al. present the results of time resolved optical measurements performed upon Co₃Sn₂S₂ crystals. They record both the time resolved reflectivity and Kerr rotation, and the most significant finding is that for a certain range of pump fluence and sample temperature, an enhancement of the magnetization may be observed. The electronic band structure is calculated with the VASP package, to confirm that Co₃Sn₂S₂ is a magnetic type-II Weyl semimetal (MWSM). The transient enhancement of the magnetization is attributed to spin-filtering at some of the observed Weyl points. This interpretation is supported by the observed temperature dependence of the enhancement, which is only observed at temperatures below 110K in the experiments, before the annihilation of the Weyl points at 0.25 eV above the Fermi level that occurs in the simulations at this temperature.

The results may extend understanding of the MWSM and offer a new mechanism for the manipulation of spin dynamics in quasi-2D van der Waals bonded materials, which is likely to be of interest to researchers studying both 2D materials and magnetism and spintronics. However, the interpretation makes many assumptions and seems rather speculative. The following issues also need to be addressed.

Response to comment:

We are grateful for the reviewer's careful review and his/her positive comments. We have revised our manuscript to address the reviewer's comments and suggestions.

Comment #1:

Few details are given of the $\text{Co}_3\text{Sn}_2\text{S}_2$ crystals. What is their size and shape, and what characterisation has been performed of structure and composition?

Response to comment #1:

We thank the reviewer's suggestions. The XRD, EDS and photo image has been added in Fig. S1, showing the high-quality of our $\text{Co}_3\text{Sn}_2\text{S}_2$ crystal.

Revision in response to comment #1:

The sentence “X-ray diffraction (XRD) and energy-dispersive X-ray spectroscopy (EDS) results (Fig. S1) show the high-quality of our sample.” has been added in the Methods section on page 16 in the main text.

Figure S1 and the caption are added in the Supplementary Information.

Figure S1 | XRD and EDS measurements of $\text{Co}_3\text{Sn}_2\text{S}_2$ crystal. a, The measured XRD result of the $\text{Co}_3\text{Sn}_2\text{S}_2$ crystal showing sharp (001) and (002) peaks. Inset is the photo image of the sample

b, The measured EDS result of the $\text{Co}_3\text{Sn}_2\text{S}_2$ crystal showing the composition of Co:Sn:S is about 3:2:2.

Comment #2:

The explanation of the time resolved optical response depends upon electrons being excited above the Fermi level by a pump beam of 800 nm (photon energy 1.55 eV). Some indication should be given of the electronic transitions induced by the pump beam. While it may be difficult to tune the wavelength of the beam, it should be possible to refer to published optical spectra and the calculated band structure.

Response to comment #2:

We appreciate the reviewer for providing this valuable suggestion. We agree with the reviewer that the optical spectra can provide significant insights into electronic transitions under pump beam excitation. Unfortunately, direct optical spectra measurements on $\text{Co}_3\text{Sn}_2\text{S}_2$ are limited. To our knowledge, the most relevant published work is Phys. Rev. Lett. 124, 077403 (2020), which we have cited in our manuscript. Although the optical spectra in that study were meticulously measured and supported by first-principle calculations, the highest applied photon energy is 1 eV, much lower than the photon energy of 1.55 eV used here. Nonetheless, we can refer to this work for some information on electronic transitions.

As depicted in Fig. 3(c) and Fig. S7 in Phys. Rev. Lett. 124, 077403 (2020), while the excitation light energy is increasing, the potential for the corresponding interband transitions also grows. Therefore, by applying pump light with a photon energy of 1.55 eV, more bands would be excited

across a broader k-space. The excited electrons are far above the Fermi level and much higher than the Weyl nodes in the momentum space. This picture is in consistent with that depicted in Phys. Rev. B 104, L100301 (2021) (Page 7 and 8 in the Supplemental Material).

Revision in response to comment #2:

The following section is included in the Supplementary Information.

S4 Electronic transition excited by the pump light

The optical spectra can provide significant insights into electronic transitions under pump beam excitation. Unfortunately, direct optical spectra measurements on $\text{Co}_3\text{Sn}_2\text{S}_2$ are limited. To our knowledge, the most relevant published work is Ref [14]. Although the optical spectra in that study were meticulously measured and supported by first-principle calculations, the highest applied photon energy is 1 eV, much lower than the photon energy of 1.55 eV used here. Nonetheless, we can refer to this work for some information on electronic transitions.

As depicted in Fig. 3(c) and Fig. S7 in Ref [14], while the excitation light energy is increasing, the potential for the corresponding interband transitions also grows. Therefore, by applying pump light with a photon energy of 1.55 eV, more bands would be excited across a broader k-space. The excited electrons are far above the Fermi level and much higher than the Weyl nodes in the momentum space. This picture is in consistent with that depicted in Ref [15].

Comment #3:

More justification should be given for the claimed bottleneck effect. The opening of a gap at the

Weyl point means that finite momentum is required for scattering across the gap. Is this the source of the bottleneck and how does the scattering rate scale with the size of the gap if electron-electron scattering is the dominant scattering mechanism?

Response to comment #3:

We thank reviewer's referee for the valuable comment.

The interband transition usually happens between the minimum of upper bands and the maximum of lower bands. As the opening of Weyl node leads to a direct gap in $\text{Co}_3\text{Sn}_2\text{S}_2$, the corresponding interband transition is the direct transition which does not require a finite momentum for scattering across the band gap.

Comment #4:

The authors should be more explicit about the nature of the half-metallic slow demagnetization mechanism. I assume they mean that the minority density of states at the Fermi level increases, allowing increased relaxation of majority spins. Please confirm or explain otherwise.

Response to comment #4:

We thank the reviewer for providing us the opportunity for clarify this point. In comparison to ferromagnetic metals, studies on magnetic half-metals [Phys. Rev. B 74, 064414 (2006), Phys. Rev. Lett. 94, 087202 (2005), Phys. Rev. X 2, 041008 (2012)] have demonstrated a distinct long demagnetization time scale which could be of up to 1000 picoseconds (ps).[Nature Mater. 8, 56

(2009)] The prominent characteristic of half-metal is that there is a spin-polarized band gap at the Fermi level so that a high spin polarization is presented. Compared to transition metals, it is proposed that this will considerably slow the demagnetization process in half-metals arising from the blocking Elliot-Yafet scattering process due to intrinsic half-metallic nature. The Elliot-Yafet spin-flip provides an interaction channel between the electron and spin systems via the band mixing for majority and minority spins. Instead of this direct angular momentum transfer, in half-metals the spin-lattice interaction plays an overwhelming role which is mediated by spin-orbit coupling [Phys. Rev. Lett. 94, 087202 (2005), Phys. Rev. B 53, 3422 (1996)] and the time constant of this spin thermalization process is much longer than the normal demagnetization time in transition metals.

Revision in response to comment #4:

The following section is included in the Supplementary Information.

S5 Long demagnetization time in magnetic half-metals

In comparison to ferromagnetic metals, studies on magnetic half-metals¹⁶⁻¹⁸ have demonstrated a distinct long demagnetization time scale which could be of up to 1000 picoseconds (ps).¹⁹ The prominent characteristic of half-metal is that there is a spin-polarized band gap at the Fermi level so that a high spin polarization is presented. Compared to transition metals, it is proposed that this will considerably slow the demagnetization process in half-metals arising from the blocking Elliot-Yafet scattering process due to intrinsic half-metallic nature. The Elliot-Yafet spin-flip provides an interaction channel between the electron and spin systems via the band mixing for majority and minority spins. Instead of this direct angular momentum transfer, in half-metals the spin-lattice interaction plays an overwhelming role which is mediated by spin-orbit coupling^{17,20} and the time

constant of this spin thermalization process is much longer than the normal demagnetization time in transition metals.

Comment #5:

Time resolved optical data is frequently discussed in the context of a two or three temperature model. Is this a useful approach in the present case?

Response to comment #5:

The reviewer has raised a very good question. Two-temperature model (involving electron and phonon temperatures) or three-temperature models (involving electron, phonon and spin temperatures) have been successfully applied in numerous studies analyzing time resolved results. However, when laser-induced magnetization dynamics involve multiple processes, the three-temperature method becomes less helpful, for example [Phys. Rev. Lett. 94, 087202 (2005), Nature Commun. 6, 6724 (2015), Adv. Mater. 31, 1806443 (2019)]. In our manuscript, if we apply an equivalent temperature for the spin system, the magnetization enhancement process would correspond to a decreasing spin temperature. This would be difficult to explain solely by considering the interactions between spin, electron, and lattice. Additional non-thermal effects have to play a significant role in the magnetization dynamics of $\text{Co}_3\text{Sn}_2\text{S}_2$. Therefore, we applied phenomenological numerical fitting combined with systematic pump fluence-, temperature-dependent measurements.

Comment #6:

The reliability of the fitting of the magneto-optical data is unclear. Fits of data acquired at a temperature of 10 K are presented in which τ_{enh} , τ_{fast} , and τ_{slow} are fixed, but the associated recovery relaxation times τ_{r1} , τ_{r2} , and τ_{r3} are allowed to vary. Table S2 shows that the fitted recovery times exhibit significant variation as the pump fluence is varied and may compensate for variations in τ_{enh} , τ_{fast} , and τ_{slow} . Therefore the assumption of fixed τ_{enh} , τ_{fast} , and τ_{slow} requires further justification.

Response to comment #6:

We understand the reviewer's concern. Firstly, we need to justify the necessity of applying numerical fitting involving three processes (excluding the initial peak). As shown in Fig. 1(j), at a low temperature of 10 K, magnetization dynamic with a very small pump fluence of 0.14 mJ/cm² is clearly dominated by fast demagnetization, whereas magnetization dynamic excited by a high pump fluence of 2.26 mJ/cm² is dominated by slow demagnetization. These two processes are easily distinguishable. Furthermore, when the pump fluence ranges from 0.28 to 1.13 mJ/cm², a significant magnetization enhancement is observed. This enhancement process cannot be included in either the fast or the slow demagnetization processes. Temperature-dependent measurements, as shown in Fig. 2(a), also demonstrate the multiple magnetization dynamics and temperature-dependent transitions. Therefore, an interesting magnetization dynamics involving three processes is justified.

The subsequent key question is how to quantitatively distinguish these processes. In fact, this fitting process needs to be approached with great care. As the reviewer pointed out, each dynamic

has two characteristic times, and the fitting results could be somewhat artificial. This is why we have presented the data fitting process in as much detail as possible in the Supplementary Information. The purpose of using fixed τ_{enh} , τ_{fast} , and τ_{slow} is to clearly distinguish these three processes. Similar methods were applied in [Phys. Rev. Lett. 121, 077204 (2018), Sci. Adv. 4, eaap9744 (2018)], which have been clarified in Section 2 of the Supplementary Information. Importantly, our temperature-dependent measurements are key to demonstrating the correlation between the Wyle nodes and the magnetization enhancement. When fitting the temperature-dependent data in Fig. 2(a), all the parameters are open to fit. We apologize for the mistake in the sentence “Therefore, we fixed several parameters correspondingly when processing the pump fluence-dependent results at 10 K and the temperature-dependent results at a fixed pump fluence.” which may have caused confusion.

Revision in response to comment #6:

The Sentence in Page 4 in Supplementary Information “Therefore, we fixed several parameters correspondingly when processing the pump fluence-dependent results at 10 K and the temperature-dependent results at a fixed pump fluence.” has been replaced by “Therefore, we fixed several parameters correspondingly when processing the pump fluence-dependent results at 10 K.”

Comment #7:

The temperatures used to label the VASP calculations are confusing at present. My impression is that the calculations are performed at zero temperature, but the authors are able to fix the moment on the Co ions. Therefore, they use the measured SQUID data in Figure S10 to determine the

moment corresponding to each temperature, and then assume that this will produce the correct band structure for that temperature. Please either confirm or explain otherwise. And what is the justification for such an approach?

Response to comment #7:

We would like to thank the reviewer for raising this important issue. In the manuscript, we used two types of temperatures. The first one, as pointed out by the reviewer, is the sample temperature at the measurement stage, as shown in Fig. 1 and Fig.2. The second one is the equivalent temperature used in the first-principle calculations. As described in the methods section, the band structures were calculated with varied fixed magnetic moment. The equivalent temperature was then calibrated using the M-T curve. This method is helpful in understanding the temperature (magnetic moment) – dependent band structures, similar to the approach used in Phys. Rev. Lett. 124, 077403 (2020) and Phys. Rev. B 104, L100301 (2021).

Revision in response to comment #7:

The equivalent temperature for first-principle calculations is explicitly pointed out in the revised manuscript. Revisions are shown in response to Comment #1 of Reviewer #1

Fig 3 has been replaced as shown in revision in response to Comment #1 of Reviewer #1

Comment #8:

I think by “electrons transmit the Weyl nodes” that authors actually mean “electrons transit

the Weyl nodes ” , but please confirm.

Response to comment #8:

We thank the reviewer's for pointing this error out. We have corrected the words.

Comment #9:

In figures S7 and S8 the captions refer to fitted curves but I don ' t see any curves in the plots.

Response to comment #9:

We are sorry for the mistakes. The curves in Fig. S7 and S8 (Fig. S8 and Fig. S9 in the revised version) are not fitted. The figure captions have been corrected.

Comment #10:

The specular optical Kerr effect (SOKE) vanishes when pump and probe polarization are the same, or if the pump has circular polarization, but is maximised when the planes of polarization of pump and probe lie 45° apart. The authors talk about SOKE in the Supplementary Material, so did they also measure the SOKE with planes of polarization of pump and probe lie 45° apart? To fully understand the zero delay peak it is also necessary to measure the transient Kerr ellipticity. Were such measurements made?

Response to comment #10:

We appreciate the reviewer for raising this important issue. We measured the initial peaks by rotating the linear polarization direction of the pump beam. The probe beam is fixed as p-polarized, and the pump beam was tuned from s polarized (0 degree), 45 degree away from s-polarized (45 degree), to p polarized (90 degree). The Kerr signals measured with positive magnetization and negative magnetization, denoted as $\Delta\theta_K(\tau, \mathbf{M})$ and $\Delta\theta_K(\tau, -\mathbf{M})$ respectively, are shown in Fig. R2 (a) and (b). The initial peaks around zero delay are highly dependent on the pump beam polarization direction. They are maximized when the planes of polarization of pump and probe are 45 degree apart. Therefore, as stated in Section 1 of the supplementary information, these peaks are attributed to SIFE/SOKE. The transient Kerr rotation changes are obtained as the asymmetric part of the measured Kerr signal with opposite external fields: $\Delta\theta_K(\tau) = (\Delta\theta_K(\tau, \mathbf{M}) - \Delta\theta_K(\tau, -\mathbf{M}))/2$, as shown in Fig. R2(c). The initial peaks, referred to as the A₁ part in the manuscript, remain unchanged with respect to the pump beam polarization direction. We also compared the initial peak in Kerr rotation with the differential curve of the Kerr signals when the pump beam is s-polarized and p-polarized under the same external field: $\Delta\theta_K(\tau, \mathbf{M}, \mathbf{0}^\circ) - \Delta\theta_K(\tau, \mathbf{M}, \mathbf{90}^\circ)$, as shown in the red line Fig. R2(d). It is evident that the A₁ peak of the Kerr rotation (black line in Fig. R1(d)) is significantly delayed compared to the SIFE/SOKE peak, with a delay time of ~ 130 fs. This observation is consistent with Fig. S6(b) in the supplementary information. Therefore, the pump beam polarization-dependent measurements support our analysis of the initial peak in the manuscript. We concur with the reviewer that ellipticity measurements may provide more information about the initial peak, which certainly deserve further study in future work.

Figure R2 a, The measured transient MOKE signal with different pump beam polarization direction under positive applied magnetic field H^+ . **b**, The measured transient MOKE signal with different pump beam polarization direction under positive applied magnetic field H^- . **c**, The transient Kerr rotation curves with different pump beam polarization direction **d**, the differential curve (red) of the Kerr signals for the pump beam is s and p polarized under the same external field $\Delta\theta_K(\tau, M, 0^\circ) - \Delta\theta_K(\tau, M, 90^\circ)$ and the transient Kerr rotation curve (black).

REVIEWERS' COMMENTS

Reviewer #1 (Remarks to the Author):

I read the manuscript and the comments by the referees. The authors have made the revision accordingly.

The authors observed intriguing properties of the magnetic Weyl semimetal. Though the analysis was rather phenomenological, the interpretation was based on the current understanding of the spin dynamics in materials. Supported by theoretical calculation, the model, proposed by the authors, likely becomes useful in the following research on magnetic Weyl semimetal.

I have noticed that there were quite a number of mistakes in the original manuscript, as pointed out by the referees. The authors should check the paper carefully again.

Reviewer #2 (Remarks to the Author):

The author addressed all my concerns and I am happy to recommend the paper for publication

Reviewer #3 (Remarks to the Author):

The authors have responded to the comments in my original report. Below I provide a brief evaluation of their responses, referring to them by the numbers used in their response and my original report.

1. Fine – the additional information is appreciated.
2. From the authors' response I conclude that no further information is available about the active optical transitions. However, given that the Weyl nodes responsible for the spin-filtering are believed to lie just 0.25 eV above the Fermi level, it is a reasonable assumption that many electrons will be optically excited to levels above these Weyl nodes.

3. The authors respond that the Weyl node leads to a direct gap. However, Fig 1f suggests that the gap is indirect. Therefore, I suggest that the authors modify the figure so as to avoid further confusion.
4. I continue to find the authors' explanation unclear. The final sentence of their response cites spin-orbit coupling but I was hoping for a simple intuitive explanation of how electrons relax from the fully spin-polarized states just above the Fermi level.
5. I suspect that a modified $2/3$ temperature model, with an additional term to account for the spin-filtering effect, could be useful but is not essential.
6. I remain concerned about the reliability of the fitting of the data at 10K, but the authors have now stated their assumptions accurately.
7. As far as I can tell from the authors' response, the temperature dependence of the calculated band structure arises from them adjusting the size of the assumed Co moment. It would be useful to the reader if they could confirm this within the manuscript.
8. Ok
9. Ok
10. The additional explanation is appreciated and I suggest that it should be included within the Supplementary Material.

The authors should make the final clarifications mentioned above, after which I believe the paper is suitable for publication.

Response to Reviewers' Comments

Response to the comments of Reviewer #1

Introductory comment:

I read the manuscript and the comments by the referees. The authors have made the revision accordingly.

The authors observed intriguing properties of the magnetic Weyl semimetal. Though the analysis was rather phenomenological, the interpretation was based on the current understanding of the spin dynamics in materials. Supported by theoretical calculation, the model, proposed by the authors, likely becomes useful in the following research on magnetic Weyl semimetal.

I have noticed that there were quite a number of mistakes in the original manuscript, as pointed out by the referees. The authors should check the paper carefully again.

Response to comment:

We thank the reviewer again for careful review and his/her positive comments. We have carefully revised our manuscript to address all the reviewers' comments and suggestions.

Response to the comments of Reviewer #2

Introductory comment:

The author addressed all my concerns and I am happy to recommend the paper for publication

Response to comment:

We thank the reviewer again for careful review and his/her positive comments.

Response to the comments of Reviewer #3

Introductory comment:

The authors have responded to the comments in my original report. Below I provide a brief evaluation of their responses, referring to them by the numbers used in their response and my original report.

Response to comment:

We are grateful for the reviewer's careful review and his/her positive comments. We have revised our manuscript to address the reviewer's comments and suggestions.

Comment #1:

- 1. Fine - the additional information is appreciated.*
- 2. From the authors' response I conclude that no further information is available about the active optical transitions. However, given that the Weyl nodes responsible for the spin-filtering are believed to lie just 0.25 eV above the Fermi level, it is a reasonable assumption that many electrons will be optically excited to levels above these Weyl nodes.*
- 3. The authors respond that the Weyl node leads to a direct gap. However, Fig 1f suggests that the gap is indirect. Therefore, I suggest that the authors modify the figure so as to avoid further confusion.*

Response to comment #1:

We thank the reviewer for pointing out this mistake, Fig. 1f has been revised.

Comment #2:

4. I continue to find the authors' explanation unclear. The final sentence of their response cites spin-orbit coupling but I was hoping for a simple intuitive explanation of how electrons relax from the fully spin-polarized states just above the Fermi level.

Response to comment #2:

We appreciate the reviewer's comment. We note that the demagnetization process (reported in the cited papers and in our manuscript) in a half-metal is a thermal effect, which does not involve the electron relaxation from the excited states. For transition magnetic metal, there are both spin-up and spin-down DOS at Fermi level. The spin-flipping process (between spin-up and spin-down electrons) via EY scattering occurs upon laser stimulation. For half metal, only spin-up or spin-down DOS exist at Fermi level, therefore the spin-flipping process is blocked. In this case, only the spin-lattice interaction play a key role and this interaction is comparably weak, resulting in a slow demagnetization. We hope this simple picture could be helpful for better understanding.

Comment #3:

5. I suspect that a modified 2/3 temperature model, with an additional term to account for the spin-filtering effect, could be useful but is not essential.

6. I remain concerned about the reliability of the fitting of the data at 10K, but the authors have

now stated their assumptions accurately.

Response to comment #3:

We appreciate the reviewer's comments.

Comment #4:

7. As far as I can tell from the authors' response, the temperature dependence of the calculated band structure arises from them adjusting the size of the assumed Co moment. It would be useful to the reader if they could confirm this within the manuscript.

8. Ok

9. Ok

Response to comment #4:

We thank reviewer for the valuable suggestion. We believe this theoretical calculation method is clarified explicitly both in the results part and the methods part in our manuscript.

Comment #5:

10. The additional explanation is appreciated and I suggest that it should be included within the Supplementary Material.

Response to comment #5:

We thank the reviewer's suggestion and this part is added in the revised Supplementary Materials.

Revision in response to comment #5:

The following section and Figure are included in the Supplementary Information.

S6 Pump beam polarization-dependent measurements of the initial peak in time-resolved magneto-optic Kerr rotation

We measured the initial peaks by rotating the linear polarization direction of the pump beam. The probe beam is fixed as p-polarized, and the pump beam was tuned from s polarized (0 degree), 45 degree away from s-polarized (45 degree), to p polarized (90 degree). The Kerr signals measured with positive magnetization and negative magnetization, denoted as $\Delta\theta_K(\tau, \mathbf{M})$ and $\Delta\theta_K(\tau, -\mathbf{M})$ respectively, **are shown in Fig. S18** (a) and (b). The initial peaks around zero delay are highly dependent on the pump beam polarization direction. They are maximized when the planes of polarization of pump and probe are 45 degree apart. Therefore, as stated in Section 1 of the supplementary information, these peaks are attributed to SIFE/SOKE. The transient Kerr rotation changes are obtained as the asymmetric part of the measured Kerr signal with opposite external fields: $\Delta\theta_K(\tau) = (\Delta\theta_K(\tau, \mathbf{M}) - \Delta\theta_K(\tau, -\mathbf{M}))/2$, as shown in Fig. R2(c). The initial peaks, referred to as the A_1 part in the manuscript, remain unchanged with respect to the pump beam polarization direction. We also compared the initial peak in Kerr rotation with the differential curve of the Kerr signals when the pump beam is s-polarized and p-polarized under the same external field: $\Delta\theta_K(\tau, \mathbf{M}, 0^\circ) - \Delta\theta_K(\tau, \mathbf{M}, 90^\circ)$, as shown **in the red line Fig. S18(d)**. It is evident that the A_1 peak of the Kerr rotation (**black line in Fig. S18(d)**) is significantly delayed compared to the SIFE/SOKE peak, with a delay time of ~ 130 fs. This observation is consistent with Fig. S6(b) in the supplementary information. Therefore, the pump beam polarization-dependent measurements support our analysis of the initial peak in the manuscript. We **note** that ellipticity measurements

may provide more information about the initial peak, which certainly deserve further study in future work.

Figure S18 a, The measured transient MOKE signal with different pump beam polarization direction under positive applied magnetic field H^+ . **b**, The measured transient MOKE signal with different pump beam polarization direction under positive applied magnetic field H^- . **c**, The transient Kerr rotation curves with different pump beam polarization direction **d**, the differential curve (red) of the Kerr signals for the pump beam is s and p polarized under the same external field $\Delta\theta_K(\tau, M, 0^\circ) - \Delta\theta_K(\tau, M, 90^\circ)$ and the transient Kerr rotation curve (black).

Comment #6:

The authors should make the final clarifications mentioned above, after which I believe the paper is suitable for publication.

Response to comment #6:

Again, we thank the reviewer for the time and effort.